# Gene Expression Space Shapes the Bioprocess Trade-Offs among Titer, Yield and Productivity

Fernando N. Santos-Navarro [1], Yadira Boada [1,2], Alejandro Vignoni [1] and Jesús Picó [1,*]

1 Synthetic Biology and Biosystems Control Laboratory, Institut d'Automática e Informática Industrial, Universitat Politècnica de Valencia, Camino de Vera s/n, 46022 Valencia, Spain; fersann1@upv.es (F.N.S.-N.); yaboa@upv.es (Y.B.); vignoni@isa.upv.es (A.V.)
2 Centro Universitario EDEM, Escuela de Empresarios, La Marina de València, Muelle de la Aduana s/n, 46024 Valencia, Spain
* Correspondence: jpico@ai2.upv.es

**Abstract:** Optimal gene expression is central for the development of both bacterial expression systems for heterologous protein production, and microbial cell factories for industrial metabolite production. Our goal is to fulfill industry-level overproduction demands optimally, as measured by the following key performance metrics: titer, productivity rate, and yield (TRY). Here we use a multiscale model incorporating the dynamics of (i) the cell population in the bioreactor, (ii) the substrate uptake and (iii) the interaction between the cell host and expression of the protein of interest. Our model predicts cell growth rate and cell mass distribution between enzymes of interest and host enzymes as a function of substrate uptake and the following main lab-accessible gene expression-related characteristics: promoter strength, gene copy number and ribosome binding site strength. We evaluated the differential roles of gene transcription and translation in shaping TRY trade-offs for a wide range of expression levels and the sensitivity of the TRY space to variations in substrate availability. Our results show that, at low expression levels, gene transcription mainly defined TRY, and gene translation had a limited effect; whereas, at high expression levels, TRY depended on the product of both, in agreement with experiments in the literature.

**Keywords:** metabolic synthetic engineering; host-aware models; cell burden; bioreactor production; multi-scale models; titer-yield-rate; gene expression; RBS strength; promoter strength



## 1. Introduction

Optimal gene expression is central to the development of both bacterial expression systems for production of heterologous proteins and microbial cell factories to produce metabolites of industrial interest. Both applications seek to obtain high levels of products of interest by means of metabolic engineering. The goal is to reach industry-level overproduction demands in an optimal way, as measured using the key performance indices titer, productivity (rate) and yield (henceforth TRY).

In practice, some trade-offs in the TRY space must be reached and be adaptive to the growth and environmental dynamic changes that occur in a bioreactor set-up [1,2]. Indeed, biomass growth and product yields cannot be simultaneously maximized. For a given substrate uptake rate, a higher growth yield will lead to a higher growth rate at the expense of the product yield.

In silico constraint-based metabolic genome-scale models have proved very valuable in engineering biosynthetic metabolic pathways in which the combined catalytic activity of a collection of enzymes is coordinated so as to produce the desired metabolites. These models provide predictions on maximum theoretical yields, optimal flux distribution to maximize flux towards some metabolite reaction bottlenecks and the required gene expression leading to increases in fluxes towards the final products [3–5]. It is possible to deal with the trade-off between yield and productivity using dynamic flux balance analysis

(dFBA), which incorporates both process dynamics and the constraint-based metabolic network model and relies on dynamic optimization methods [6–8].

In practice, the metabolic costs of producing proteins, enzymes, and other cell macromolecules during gene over-expression often lead to shifts away from the optimal predictions. Yet, the focus of these constraint-based models on metabolism excludes proteins and other macromolecules, the major biomass constituents, from the model. Models that integrate metabolism, biomass composition and the cost for the cell to produce a certain protein or enzyme can potentially yield better predictions than those focused on metabolism in isolation [9]. Thus, for instance, dynamic enzyme-cost FBA [10,11] includes dynamic changes in substrate concentration, a detailed description of biomass composition, and takes into account how much it costs to the cell to produce the desired enzyme.

As an alternative to metabolism-centered models, in the recent years there has been a increasing interest in the development of models of gene expression accounting for cellular resources competition [12,13]. These range from very coarse-grain ones [14–17] to semi-mechanistic ones with varied degrees of granularity [18–20]. This class of models may consider the interplay between substrates uptake, cell growth rate, gene expression and interaction with the cell host caused by competition for cellular resources. Therefore, they have the potential to deal with the related issues of dynamic gene regulation, cell resource allocation and the varying industry-scale bioreactor environment.

As an intermediate modeling strategy, whole-cell models connecting gene expression, metabolism and growth, such as the coarse-grained self-replicator models, take into account the dynamic feedback from gene expression and growth to metabolism [21,22].

So far, the main interest in the literature has been in determining the required optimal gene expression levels that fulfill the design specifications using any of the modeling approaches described above. The dynamic regulation of the the specified expression levels has also been addressed both for heterologous protein expression systems [23] and for metabolic pathways [24]. Indeed, major improvements in yield, titer and productivity of engineered metabolic pathways can be accomplished by dynamic balancing of pathway gene expression [25,26], where the application of dynamic feedback and feedforward regulation of gene expression addresses the robustness pitfalls of static regulation.

It is known that weakly expressed endogenous genes exhibit low RNA polymerase/ribosome ratios, while strongly expressed genes have higher RNAP/ribosome ratios, as this is metabolically efficient [16]. However, to the author's knowledge, little research has been done about the differential roles that gene transcription and translation can play in shaping the trade-offs between titer, yield and productivity at the bioreactor level.

Here, we use a multi-scale model that connects the dynamics of the population of cells in the bioreactor, those of substrate uptake with the dynamic interaction between the host and the synthetic circuits expressing proteins of interest. Our model predicts the cell growth rate and the distribution of cell mass between the protein of interest and the host ones as a function of the substrate uptake and the main lab-accessible gene expression-related characteristics: promoter strength, gene copy number and ribosome binding site (RBS) strength. While in this work we do not consider metabolism explicitly, we do it implicitly through the substrate dynamics. For a wide range of gene expression levels, we evaluate the differential roles of gene transcription and translation in shaping the trade-offs between titer, yield and productivity rate for the three main operational modes of industrial bioreactors. We also evaluate the sensitivity of the mapping between the expression and the TRY spaces as a function of variations in the substrate availability.

## 2. Materials and Methods

In this section the mathematical models and methodological elements used in this work are presented. First we introduce our multi-scale model including the bioreactor model, the host model and the synthetic circuit model. Then we dive into the synthetic circuit gene expression space and into the bioreactor modes of operation. Finally, we present the TRY performance indices and their relative variation indices for substrate variations.

### 2.1. Multi-Scale Model

In order to take into account the different scales involved in the model of a bioprocess, we consider a multi-scale model integrating the different occurring phenomena. From top to bottom (Figure 1A), our model describes the interactions in the three levels considered: the biomass and substrate dynamics in the bioreactor, the dynamics of the host and those of the synthetic gene circuit expressing the proteins of interest.

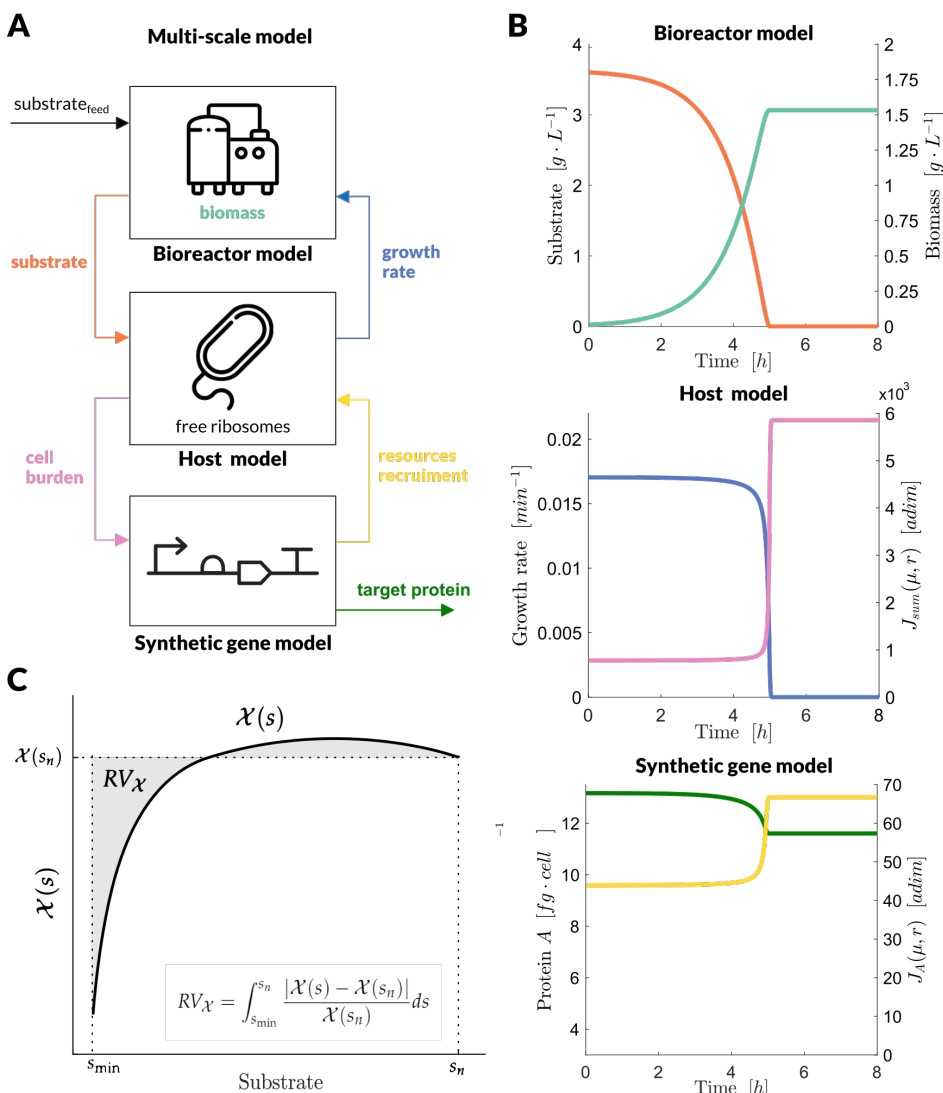

**Figure 1.** Bioprocess multi-scale model. (**A**). Schematic representation of the multi-scale model. The three levels considered including the bioreactor, the host cell and the synthetic circuit models and their interactions. (**B**). Simulation obtained with the multi-scale model. The colors correspond to the interactions in panel (A). An example of a batch operation, where the synthetic circuit parameters are $\omega_A = 50$, $k_b^A = 5$ and $K_u^A = 117$. Top panel shows the macroscopic scale with the substrate availability (left axis) and biomass concentration (right axis). Middle panel shows the growth rate (left axis) and a measure of the cell burden (right axis). Bottom panel shows the amount of protein A expressed in a cell (left axis) and its resource recruitment strength $J_A$ (right axis). (**C**). TRY relative variation indices for substrate variations. Shaded area $RV_{\mathcal{X}}$ represents the integral from minimum substrate $s_{min}$ to the nominal substrate $s_n$ of the difference between the nominal index value and the actual value.

At the top of our multi-scale model we find the macroscopic dynamics of the substrate and biomass in the bioreactor (Figure 1A, Bioreactor model). This bioreactor model takes as inputs the substrate inflow and outflow, together with the specific growth rate of the cell

population in the bioreactor, and it provides the concentration of available substrate and biomass in the bioreactor. These dynamics are defined by the following set of equations:

$$\dot{V} = F_{in} - F_{out} \tag{1}$$

$$\dot{n} = \mu n - \frac{F_{in}}{V} n \tag{2}$$

$$\dot{s} = \frac{F_{in}}{V}(s_{feed} - s) - y^{-1}\mu x(n, \mu) \tag{3}$$

where $V$ is the volume of culture in the bioreactor in $L$, $n$ is the concentration of cells in the bioreactor (cell $\cdot$ L$^{-1}$), and $s$ is the amount of available substrate (g $\cdot$ L$^{-1}$). $F_{in}$ and $F_{out}$ represent the input and output fluxes (L $\cdot$ min$^{-1}$), and $\mu$ the specific cell growth rate in min$^{-1}$.

At the intermediate level (Figure 1A) the host model represents the dynamics of the cell and takes as inputs the available substrate from the bioreactor model and a measure of the gene expression resources demanded by the synthetic gene circuit. This host model provides the cell-specific growth rate and a measure of the cell burden, and it is defined as follows:

$$\mu = \frac{m_{aa}}{m_p(\mu)} \nu(s) \phi_b(\mu, r, s) \phi_t r_t(m_r) \tag{4}$$

$$r = \frac{\phi_t r_t(m_r)}{1 + J_{sum}(\mu, r, s)} \tag{5}$$

$$\dot{m}_r = \left[ m_p(\mu) \frac{N_r J_r(\mu, r, s)}{J_{sum}(\mu, r, s)} - m_r \right] \mu \tag{6}$$

$$\dot{m}_{nr} = \left[ m_p(\mu) \frac{N_{nr} J_{nr}(\mu, r, s)}{J_{sum}(\mu, r, s)} - m_{nr} \right] \mu \tag{7}$$

where the main variables can be found in Table 1, and the terms $J_r(\mu, r, s)$, $J_{nr}(\mu, r, s)$ are the resources recruitment strengths for the host ribosomal and non-ribosomal lumped ensembles of genes, and $J_{sum}(\mu, r, s) = \sum_{i=r,nr,A} N_i J_i(\mu, r, s)$ is the total resources recruitment strength of the cell which represents the cell burden. These are the key functional coefficients that allow us to explain the distribution of resources in the host and the relationship between the usage of resources, cell growth and protein productivity. Details of the host–circuit interaction model derivation can be found in our previous work [27]. The parameters used in this work were fitted from experimental data from [28,29] and correspond to the wild-type *E. coli* K-12 strain MG1655 [30] and *E. coli* B/r strain.

The internal variables in the model and the values of the parameter can be found in Table 2 and Table 3 respectively.

**Table 1.** States and main variables of the host model.

| Name | Description | Units |
| --- | --- | --- |
| $r$ | Free mature ribosomes in the cell. | molec $\cdot$ cell$^{-1}$ |
| $m_r$ | Total mass of ribosomal proteins in the cell. | fg $\cdot$ cell$^{-1}$ |
| $m_{nr}$ | Total mass of non ribosomal proteins in the cell. | fg $\cdot$ cell$^{-1}$ |
| $m_A$ | Total mass of protein $A$ in the cell. | fg $\cdot$ cell$^{-1}$ |

**Table 2.** Internal variables of the model.

| Name | Description | Equation | Units |
|---|---|---|---|
| $\nu(s)$ | Effective translation rate per ribosome. | $\nu(s) = \nu_{max}s/(s + K_s)$ | $aa \cdot min^{-1}$ |
| $k_e(s)$ | Translation initation rate. | $k_e(s) = \nu(s)/l_e$ | $min^{-1}$ |
| $J_{sum}(\mu, r, s)$ | Total sum of all the J in the cell. | $J_{sum}(\mu, r, s) = \sum_{i=r,nr,A} N_i J_i(\mu, r, s)$ | adim |
| $K^r_{C_0}(s)$ | Effective RBS affinity of ribosomal mRNA. | $K^r_{C_0}(s) = k^r_b/(k^r_u + k_e(s))$ | $cell \cdot molec^{-1}$ |
| $E^r_m$ | Ribosomes density related term for ribsomal mRNA. | $E^r_m = 0.62l^r_p/l_e$ | adim |
| $J_r(\mu, r, s)$ | Average J value of one ribosomal *E. coli.* gene. | $J_r(\mu, r, s) = E^r_m \omega_r/(d^r_m/K^r_{C_0}(s) + \mu r)$ | adim |
| $r_t(m_r)$ | Number of mature and inmmature ribosomes. | $r_t(m_r) = m_r/r_w$ | $molec \cdot cell^{-1}$ |
| $\phi_b(\mu, r, s)$ | Fraction of translating ribosomes of $\phi_t r_t(m_r)$. | $\phi_b(\mu, r, s) = J_{sum}(\mu, r, s)/(1 + J_{sum}(\mu, r, s))$ | adim |
| $K^{nr}_{C_0}(s)$ | Effective RBS affinity of non-ribosomal mRNA. | $K^{nr}_{C_0} = k^{nr}_b/(k^{nr}_u + k_e(s))$ | $cell \cdot molec^{-1}$ |
| $E^{nr}_m$ | Ribosomes density related term for non-ribsomal mRNA. | $E^{nr}_m = 0.62l^{nr}_p/l_e$ | adim |
| $J_{nr}(\mu, r, s)$ | Average J value of one non-ribosomal *E. coli.* gene. | $J_{nr}(\mu, r, s) = E^{nr}_m \omega_{nr}/(d^{nr}_m/K^{nr}_{C_0}(s) + \mu r)$ | adim |
| $K^A_{C_0}(s)$ | Effective RBS affinity of protein A mRNA. | $K^A_{C_0}(s) = k^A_b/(k^A_u + k_e(s))$ | $cell \cdot molec^{-1}$ |
| $E^A_m$ | Ribosomes density related term for protein A mRNA. | $E^A_m = 0.62l^A_p/l_e$ | adim |
| $J_A(\mu, r, s)$ | Average J value of one protein A gene. | $J_A(\mu, r, s) = E^A_m \omega_A/(d^A_m/K^A_{C_0}(s) + \mu r)$ | adim |
| $m_p(\mu)$ | Total protein mass of the cell. | $m_p(\mu) = c_1\mu^2 + c_2\mu + c_3$ | $fg \cdot cell^{-1}$ |
| $x(n, \mu)$ | Concentration of biomass in the biorreactor. | $x(n, \mu) = nm_p(\mu)$ | g/L |

**Table 3.** Parameters of the model.

| Name | Description | Value | Units | Reference |
|---|---|---|---|---|
| $K_s$ | Half activation threshold of growth rate. | 0.1802 | $g \cdot L^{-1}$ | [31] |
| $\nu_{max}$ | Maximum effective translation rate per ribosome. | 1260 | $aa \cdot min^{-1}$ | [32] |
| $m_{aa}$ | Average aminoacid mass. | $182.6 \times 10^{-9}$ | $fg \cdot aa^{-1}$ | [33] |
| $l_e$ | Ribosome occupancy length. | 25 | aa | estimated [34–37] |
| $\phi_t$ | Fraction of mature available ribosomes relative to the total. | 0.7796 | adim | [17,29] * |
| $l^r_p$ | Mean length of ribosomal proteins. | 195 | aa | calculated from [28] |
| $d^r_m$ | Mean degradation rate of ribosomal mRNA. | 0.16 | $min^{-1}$ | calculated from [28] |
| $k^r_u$ | Dissotiation rate RBS-ribosome for ribosomal mRNA. | 135 | $min^{-1}$ | [27] * |

**Table 3.** *Cont.*

| Name | Description | Value | Units | Reference |
|---|---|---|---|---|
| $k_b^r$ | Association rate RBS-ribosome for ribosomal mRNA. | 8.853 | $\text{cell} \cdot \text{min}^{-1} \cdot \text{molec}^{-1}$ | [27] * |
| $N_r$ | Number of proteins that make up a ribosome. | 55 | adim | [27] |
| $\omega_r$ | Average transcription rate for ribosomal proteins. | 4.8658 | $\text{molec} \cdot \text{min}^{-1} \cdot \text{cell}^{-1}$ | [27] * |
| $r_w$ | Weight of a ribosome. | 0.0045 | fg | [29] |
| $l_p^{nr}$ | Mean length of non-ribosomal proteins. | 333 | aa | calculated from [28] |
| $d_m^{nr}$ | Mean degradation rate of non-ribosomal mRNA. | 0.2 | $\text{min}^{-1}$ | calculated from [28] |
| $k_u^{nr}$ | Dissotiation rate RBS-ribosome for non-ribosomal mRNA. | 6.1297 | $\text{min}^{-1}$ | [27] * |
| $k_b^{nr}$ | Association rate RBS-ribosome for non-ribosomal mRNA. | 14.9971 | $\text{cell} \cdot \text{min}^{-1} \cdot \text{molec}^{-1}$ | [27] * |
| $N_{nr}$ | Number of non ribosomal proteins expressed at one time. | 1735 | adim | [27] |
| $\omega_{nr}$ | Average transcription rate for non ribosomal proteins. | 0.03 | $\text{molec} \cdot \text{min}^{-1} \cdot \text{cell}^{-1}$ | [27] * |
| $l_p^A$ | Length of protein A. | 195 | aa | ** |
| $d_m^A$ | Mean degradation rate of protein A mRNA. | 0.16 | $\text{min}^{-1}$ | ** |
| $k_u^A$ | Dissotiation rate RBS-ribosome for protein A mRNA. | (6 135) † | $\text{min}^{-1}$ | [38,39] |
| $k_b^A$ | Association rate RBS-ribosome for protein A mRNA. | (3 15) † | $\text{cell} \cdot \text{min}^{-1} \cdot \text{molec}^{-1}$ | [38–40] |
| $N_A$ | Number of copies of gene $A$. | (1 70) † | adim | ** |
| $\omega_A$ | Average transcription rate for protein A. | (0 5) † | $\text{molec} \cdot \text{min}^{-1} \cdot \text{cell}^{-1}$ | ** |
| $c_1$ | First coefficient of mass equation. | 239,089 | $\text{fg} \cdot \text{cell}^{-1} \cdot \text{min}^2$ | [27] * |
| $c_2$ | Second coefficient of mass equation. | 7432 | $\text{fg} \cdot \text{cell}^{-1} \cdot \text{min}$ | [27] * |
| $c_3$ | Third coefficient of mass equation. | 37.06 | $\text{fg} \cdot \text{cell}^{-1}$ | [27] * |
| $y$ | Biomass yield on glucose. | 0.45 | $\text{g}_{\text{biomass}} \cdot \text{g}_{\text{substrate}}^{-1}$ | [41] |
| $s_{feed}$ | Fresh media substrate concentration. | 3.6 | $\text{g} \cdot \text{L}^{-1}$ | [31] |

* These parameters were re-optimized following the methods described in [27] to better fit the wild-type at low growth rate, since that range is the most relevant for this work. ** Without loss of generality in the results, we choose $l_p^A$ and $d_m^A$ to be equal to the ribosomal parameters, and the range of $N_A$ and $\omega_A$ to be in the order of ribosomal and non-ribosomal parameters. † These parameters define the gene expression space and their value varies in simulations between the minimun value (first value in parentheses) and the maximum value (last value in parentheses).

Finally, at the bottom level, a model of the synthetic gene circuits considers the interaction dynamics with the host and estimates production of the mass fraction of the proteins of interest (8). In this work, as an example, we consider the expression of a generic theoretical gene $A$. More complex synthetic circuits or metabolic pathways with their corresponding enzymes can be easily incorporated into the model by adding more equations such as (8).

$$\dot{m}_A = \left[ m_p(\mu) \frac{N_A J_A(\mu, r, s)}{J_{sum}(\mu, r, s)} - m_A \right] \mu \tag{8}$$

where $m_A$ is the total mass of protein $A$ in the cell, $N_A$ is the number of copies of gene $A$, and $J_A(\mu, r, s)$ is the resources recruitment strength of the gene $A$.

The resources recruitment strength of a given gene, gene $A$ in this case, $J_A(\mu, r, s)$ can be understood as a measure of its eagerness to capture cell resources to get expressed. As seen from Table 2, it can be expressed as:

$$J_A(\mu, r, s) = 0.62 \frac{l_p^A}{l_e} \frac{\omega_A}{\frac{d_m^A}{K_{C_0}^A(s)} + \mu r} \tag{9}$$

where $l_p^A / l_e$ is the ribosomes density (protein length over ribosomes occupancy length), the product $\mu r$ of growth rate and number of free ribosomes is the flux of free resources, and $d_m^A$ is the degradation rate of the transcript. The transcription rate is $\omega_A$, the RBS strength is $K_{C_0}^A(s)$, and the gene copy number is $N_A$. In this work we analyze the roles that gene transcription and translation may play in shaping the trade-offs between titer, yield and productivity at the bioreactor level. Thus, on the one hand we will consider the transcription rate per gene $\omega_A$ times the gene copy number $N_A$. Notice $\omega_A$ is directly related to the promoter strength. On the other hand we consider the RBS strength. The effective translation rate depends on the availability of the substrate.

In our model, the RBS strength for a gene expressing the protein $A$ is defined as:

$$K_{C_0}^A(s) = \frac{k_b^A}{k_u^A + k_e(s)} \tag{10}$$

where $k_b^A$ and $k_u^A$ are the association and dissociation rates between the ribosome binding site and the transcript, respectively, and $k_e(s)$ is a Monod-like function of the extracellular substrate $s$ that models the translation initiation rate as a function of the maximum translation rate per ribosome, the ribosomes density, the availability of substrate that can be catabolized to build aminoacids, and the affinity of the host for the substrate (see Tables 2 and 3 and reference [27]).

The fact that this RBS strength definition depends on the available substrate as in Equation (10) has several implications. First, notice that an infinite number of different combinations of $k_b^A$ and $k_u^A$ can give the same RBS strength. This is not different from the situation when considering the substrate-independent saturated RBS strength obtained for substrate saturation:

$$K_{C_0, \text{sat}}^A = \frac{k_b^A}{k_u^A + v_{\max}/l_e} \tag{11}$$

Most important, the sensitivity of the substrate-dependent RBS strength to changes in the substrate concentration in the culture depends on the actual values of $k_b^A$ and $k_u^A$. Notice that for higher values of the saturated RBS strength, the variation of the substrate-dependent RBS-strength as a function of $k_e(s)$ is larger, with:

$$\frac{1}{K_{C_0}^A(s)} \frac{\partial K_{C_0}^A(s)}{\partial k_e(s)} = -\frac{1}{k_b^A} K_{C_0}^A(s) \tag{12}$$

Therefore, in our analysis we evaluated the values of titer, productivity rate and yield for a nominal value of the substrate concentration in the culture (see Section 2.2) as a function of the expression space determined by the parameters $N_A \omega_A$, $k_b^A$ and $k_u^A$. In addition, we also evaluated the sensitivity of the TRY space to changes in the substrate concentration as a function of the expression space.

### 2.2. Bioreactor Model—Modes of Operation

The extracellular macroscopic inflow $F_{in}$ and outflow $F_{out}$ in model (1)–(3) depend on the mode of operation considered for the bioreactor: batch, fedbatch or continuous.

We assumed a well-mixed, homogeneous culture and the values of parameters defined in Table 3. The initial conditions for the concentration of cells in the bioreactor, the substrate concentration and the culture initial volume were set to industrially plausible values for a small size bioreactor with volume up to 10 L: $n(0) = 5 \times 10^{10}$ cell $\cdot$ L$^{-1}$, $s(0) = 3.6$ g $\cdot$ L$^{-1}$ and $V(0) = 1$ L, respectively. To calculate the initial conditions for the host cell parameters $\mu$, $r$, $m_r$, $m_{nr}$ and $m_A$, we ran a simulation for constant substrate $s = s(0)$, and we obtained the steady-state values for all the variables in the system. These were used as initial conditions for the host cell parameters.

For the batch mode of operation in the bioreactor, we set $F_{in} = F_{out} = 0$, and we ran the simulations until the substrate concentration in the bioreactor decreased below 2% of its initial value.

For the fedbatch and continuous modes, substrate feeding policies were applied such that the concentration of substrate in the bioreactor was kept constant to the nominal value $s_n = s(0) = 3.6$ g $\cdot$ L$^{-1}$. For the fedbatch mode we used $F_{out} = 0$ and the substrate feeding law

$$F_{in}(\mu, n, V, s) = y \mu x(\mu, n) \frac{V}{s_{feed} - s} \tag{13}$$

The feeding law (13) makes the substrate concentration in the bioreactor to remain constant and equal to the initial one [31]. For the substrate feeding concentration we used $s_{feed} = 180.156$ g $\cdot$ L$^{-1}$, and the bioreactor was fed until the culture volume reached 10 L. At this point the feeding inflow was set to zero ($F_{in} = 0$), and the simulation was continued in batch mode until the substrate concentration in the bioreactor decreased below 2% of its initial value.

Finally, for continuous mode, we used the feeding law $F_{out} = F_{in} = VD(\mu, n)$ with:

$$D(\mu, n) = \mu \frac{x(n, \mu)}{x_{ref}} \tag{14}$$

where $D(\mu, n)$ is the dilution and $x_{ref}$ a reference value for the concentration of biomass in the bioreactor. We used $x_{ref} = 1$ g $\cdot$ L$^{-1}$ to get a production comparable with batch mode. When the reference is achieved, the concentration of glucose is kept at $s = 1.4$ g $\cdot$ L$^{-1}$. However, the biomass starts in the simulations approximately at 0.01 g $\cdot$ L$^{-1}$, and it takes almost all the simulation time to get to the reference. Therefore, for most of the simulation time the substrate was close to $s_n = 3.6$ g $\cdot$ L$^{-1}$ as in fedbatch case and in the exponential phase of the batch one.

The simulation ran for a time interval equivalent to the turnover time, so that 10 L of culture was introduced and removed form the bioreactor.

Notice the conditions in the three modes of operation of the bioreactor were chosen so that the metabolic state of the cells in all three cases were equivalent so as to achieve a fair comparison.

To keep track of the volume of media and product *A* added and removed from the bioreactor we extended the model with the expressions:

$$\dot{V}_{feed} = F_{in} \tag{15}$$

$$\dot{V}_{out} = F_{out} \tag{16}$$

$$\dot{S}_{out} = F_{out}\, s \tag{17}$$

$$\dot{M}_{A_{out}} = F_{out}\, n\, m_A, \tag{18}$$

where $V_{feed}$ is the total volume of media fed to the bioreactor, $V_{out}$ the total volume of media removed from the bioreactor, $S_{out}$ the total mass of substrate removed from the bioreactor and $M_{A_{out}}$ the total mass of protein *A* removed from the bioreactor. We assumed that the substrate removed from the bioreactor due to $F_{out}$ is recovered and that cells stop growing once the substrate is removed.

### 2.3. TRY Performance Indices

Three measures are commonly used to determine the performance of a bioprocess [31]: titer *T*, volumetric productivity rate *R* and yield *Y*. We evaluated them for the production of the protein *A* in *E. coli* under the three modes of operation of the bioreactor and for different points covering the expression space (promoter and RBS strength).

The titer *T*, i.e., the concentration of the molecule of interest at the end of the bioprocess, was measured in units of grams per liter and was calculated as:

$$T = \frac{V^f n^f m_A^f + M_{A_{out}}^f}{V^f + V_{out}^f}, \tag{19}$$

where the superscript *f* indicates the final time of the fermentation, $m_A$ is the amount of protein *A* mass in one cell and *n* is the concentration of cells in the bioreactor.

The average volumetric productivity rate *R*, i.e., the production of the molecule of interest (protein *A*) per time unit, was measured in units of grams per liter per hour and calculated as:

$$R = \frac{T}{t^f}, \tag{20}$$

where $t^f$ is the final time of the fermentation.

Finally, the yield, i.e., the conversion factor of substrate into the product (protein or metabolite) of interest, was measured in units of product grams per substrate grams and was calculated as:

$$Y = \frac{T(V^f + V_{out}^f)}{s^0 V^0 - s^f V^f + s_{feed} V_{feed}^f - S_{out}^f}, \tag{21}$$

where the superscripts $\{0, f\}$ indicate the initial and final time of the fermentation process, *V* is the culture volume in the bioreactor, *s* the substrate concentration in the bioreactor and $s_{feed}$ is the substrate concentration in the feed stream.

We used the nominal substrate concentration $s_n = 3.6\,\text{g} \cdot \text{L}^{-1}$ to evaluate the nominal values of titer, productivity rate and yield as a function of the expression space determined by the parameters $N_A \omega_A$, $k_b^A$ and $k_u^A$.

### 2.4. TRY Relative Variation Indices for Substrate Variations

Fluctuations in the availability of limiting substrate are one of the main disturbances that cells may encounter within the bioreactor environment. Indeed, if the concentration of limiting substrate in the bioreactor decreases (e.g., because of mixing heterogeneity or saturation), the achieved TRY will change as the availability and distribution of cell resources vary.

To evaluate the sensitivity of the TRY space to changes in the substrate concentration in the culture as a function of the expression space, we used a measure of the relative variation of titer, rate and yield with respect to their nominal values for a range of variation in the substrate concentration, as it can be seen in Figure 1C. Thus, we defined the index:

$$RV_{\mathcal{X}} = \int_{s_{\min}}^{s_n} \frac{|\mathcal{X}(s) - \mathcal{X}(s_n)|}{\mathcal{X}(s_n)} ds \tag{22}$$

where $\mathcal{X} = \{T, R, Y\}$ for titer, productivity rate and yield, respectively. The index is the cumulative relative difference between $\mathcal{X}$ evaluated for the nominal substrate concentration and for a range $s_{\min} \leq s \leq s_n$ using Equations (19)–(21).

## 3. Results and Discussion

### 3.1. Nominal TRY as a Function of the Expression Space and Bioreactor Operation

The analysis performed with different conditions of bioreactor operation and several combinations spanning the expression space reveals, as expected, that the cell growth rate decreases as the cell burden induced by higher expression levels of $A$ increases. Figure 2 shows the results for the nominal variation of the TRY space as a function of the expression space for the three modes of operation in the bioreactor. We varied the parameters $N_A \omega_A$ and $K_{C_0}^A(s_n)$ of the expression space in a range from low to high expression of the protein $A$ (see Figure 2C).

Titer and yield show a decreasing monotonous relationship with growth rate. Thus, as observed in Figure 2A, titer and yield decrease as the cell burden decreases (i.e., weak promoter and RBS strengths are used) and, consequently, the cell growth increases. Indeed, to obtain higher titer and yield, the cell must invest a higher fraction of its mass to synthesize the protein $A$. As a result, it uses fewer cell resources to build ribosomes, and the cell growth decreases [27].

On the contrary, the maximum productivity rate depends non-monotonously on the growth rate, achieving a maximum for $\mu \approx 0.075\,\mathrm{min}^{-1}$.

Thus, we find the same qualitative trade-offs between titer, yield and rate as a function of growth rate encountered when trying to optimize the metabolic flux towards a product of interest using metabolic flux analysis [42,43]. There is a trade-off between titer and yield on the one hand, and productivity rate on the other. High titer and yield cannot be attained without eventually decreasing the productivity rate.

Additionally, as expected, the fedbatch mode achieves higher values of productivity rate and titer of the protein $A$ for all the combinations in the expression space (see Figure 2A). This simply reflects that in fedbatch mode the total amount of substrate fed to the bioreactor is larger than in batch and continuous modes. Yet, the normalized results show no relevant differences (Figure 2B). That is, the normalized titer rate and yield do not depend on the mode of operation of the bioreactor but only on the the promoter and RBS strengths in the expression space.

This result provides a principle of space-scale separation for multi-scale models. For a given substrate availability, and assuming homogeneity in the population of cells in the bioreactor, the cell growth for each individual local cell and the mass of heterologous proteins it will express depend on the interactions between the cell host and the genes, being independent of the way the substrate is fed to the population of cells. The bioreactor mode of operation will affect the total amount of substrate fed during the fermentation process and, therefore, the size of the population of cells. Although the geometry and other physical characteristics of the bioreactor are not considered here, they could be important to provide a way to characterize heterogeneous distributions of the limiting substrate within the bioreactor. Individual cells with different substrate concentrations will be subject to different interactions between the host and the synthetic circuit, and then they might synthesize the heterologous proteins at different rates.

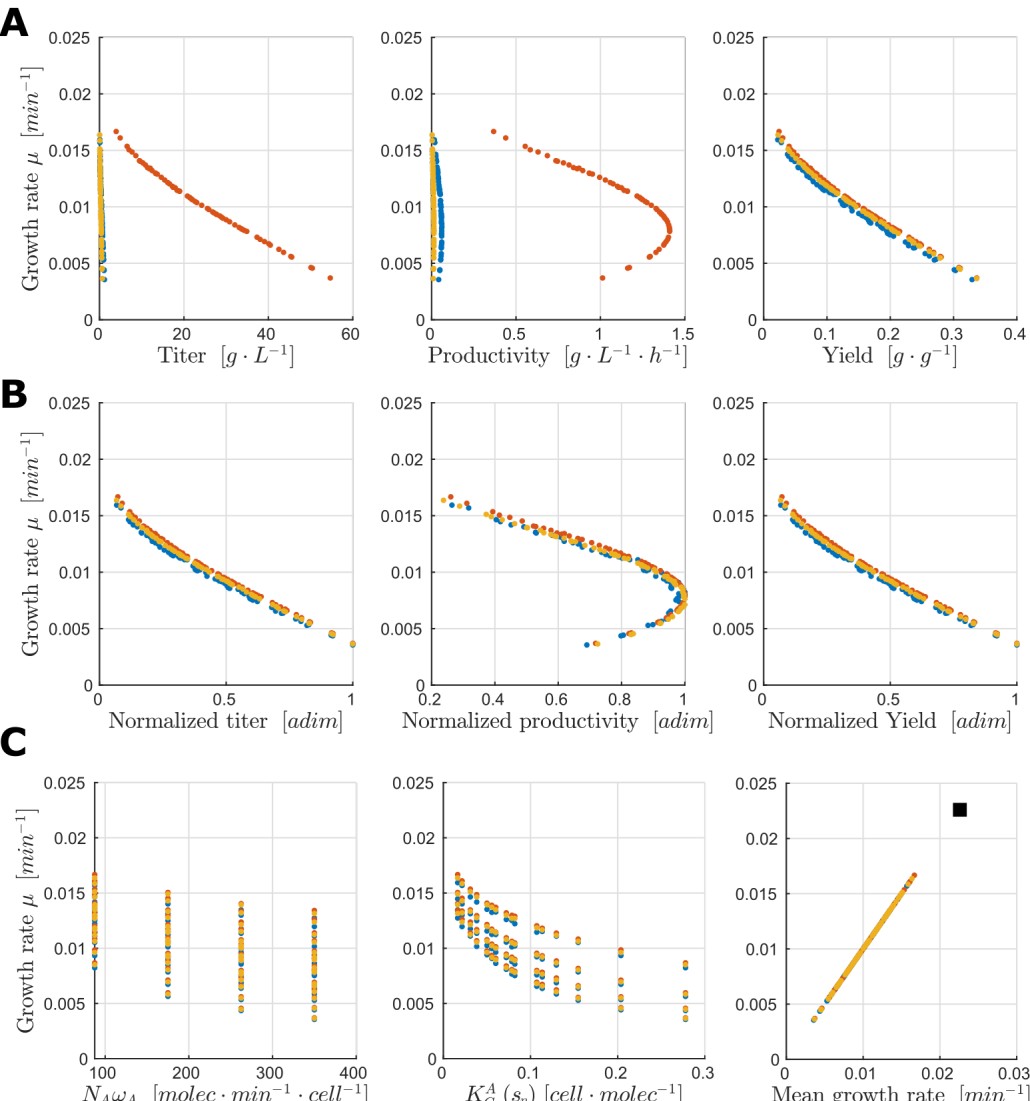

**Figure 2.** TRY of protein *A* across the gene expression space for batch, fedbatch and continuous operation obtained by simulation with the multi-scale model. All plots share the growth rate as the y-axis. Each dot corresponds to a different combination of transcription $N_A \omega_A$ and RBS $K_{C_0}^A(s_n)$ strengths. The dot color corresponds to the following: blue, batch; red, fedbatch; yellow, continuous. (**A**) Absolute titer, productivity rate and yield. (**B**) Normalized titer, productivity rate and yield. (**C**) Shows the combinations of $N_A \omega_A$ and $K_{C_0}^A(s_n)$ used and their associated induced growth rate (note that we plot $K_{C_0}^A(s_n)$ at nominal substrate $s_n = 3.6\,\text{g}\cdot\text{L}^{-1}$, since $K_{C_0}^A(s)$ is not constant because *s* changes in simulation time). The wild-type growth rate is shown with a black square (right).

## 3.2. Mapping between Gene Expression and TRY Spaces

In this section, we investigated different combinations of promoter and RBS strength and their corresponding regions in TRY space. Figure 3 shows the results of TRY space for fedbatch mode. The colored dots represent different combinations of $N_A \omega_A$ and $K_{C_0}^A(s_n)$ and TRY space, and the gray dots show the entire expression space.

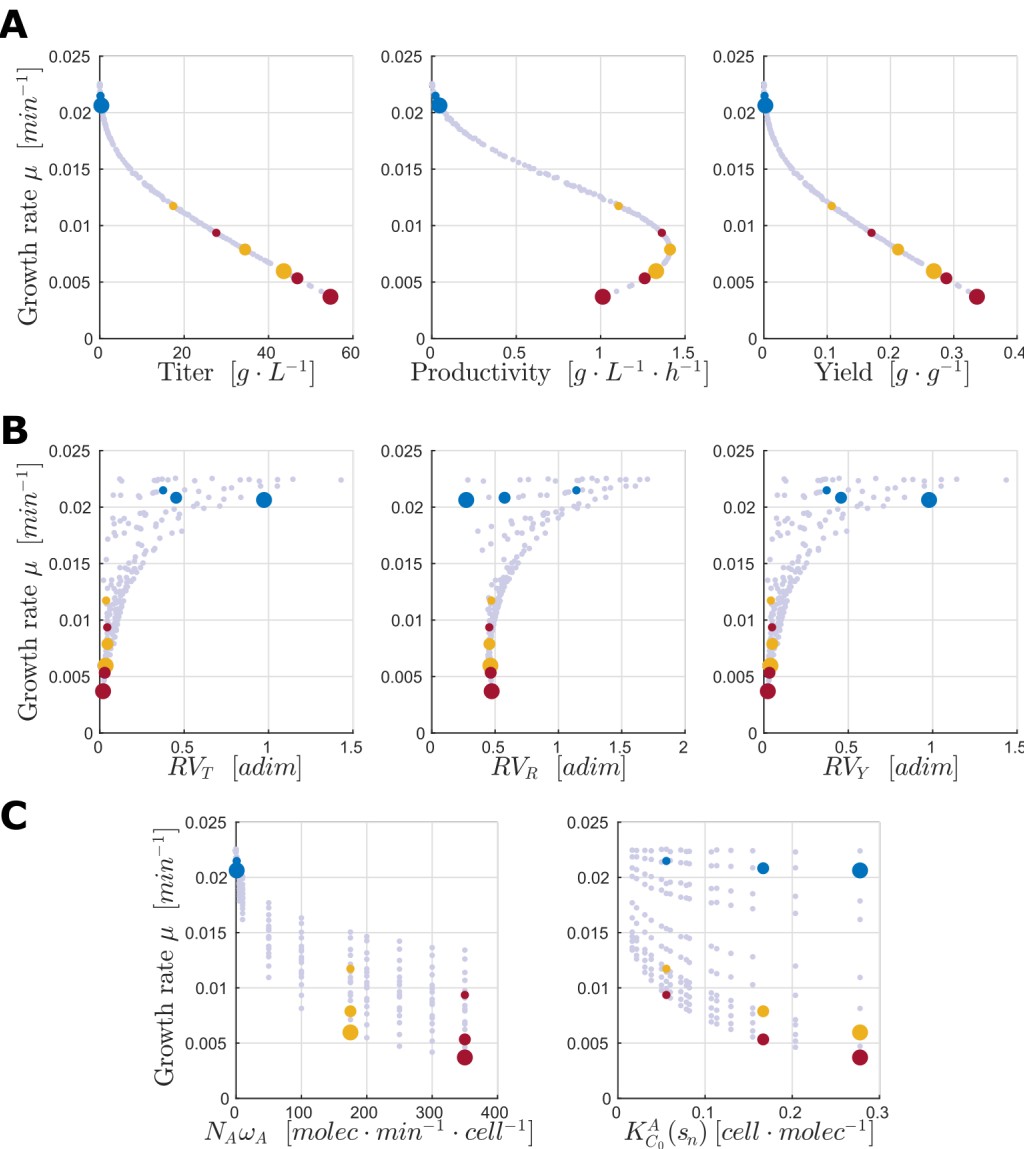

**Figure 3.** TRY values and relative variation indices of protein *A* for all the gene expression space in fed-batch operation obtained by simulation with the multi-scale model. All the plots share the growth rate as the y-axis. Each one dot corresponds to a different combination of transcription rate $N_A\omega_A$ and effective RBS $K_{C_0}^A(s_n)$ (note that we plot $K_{C_0}^A(s_n)$ at nominal substrate $s_n = 3.6\,\mathrm{g}\cdot\mathrm{L}^{-1}$, since $K_{C_0}^A(s)$ is not constant because $s$ changes in simulation time). The colors blue, yellow and red represent low, medium and high $N_A\omega_A$ respectively, while the sizes of the markers are small, medium and big and show low, medium and high $K_{C_0}^A(s_n)$. The gray color respresent the rest of $N_A\omega_A$ and $K_{C_0}^A(s_n)$ combinations. (**A**) TRY measurements: titer, productivity and yield. (**B**) TRY relative variation indices for substrate variations. (**C**) Transcription rate $N_A\omega_A$ and the effective RBS strength $K_{C_0}^A(s_n)$.

At low protein expression levels, the promoter strength mainly determines the TRY values, while the RBS strength weakly influences its value. Figure 3A shows that the low protein expression region ($\mu \in [0.015, 0.025]$ min$^{-1}$) corresponds to low values of $N_A\omega_A$ (blue dots), and the medium to high values (yellow and red dots) are outside this region. However, varying the value of $K_{C_0}^A(s_n)$ (represented in the size of the dots) hardly changes the value of the TRY space, and they remain within the region of low protein expression levels. This distinct behavior between varying $N_A\omega_A$ and $K_{C_0}^A(s_n)$ is because in (9) the value of $J_A(\mu, r, s)$ is proportional to the value of $N_A\omega_A$. Whereas the value of $K_{C_0}^A(s)$ is in the denominator of $\omega_A/(d_m^A/K_{C_0}^A(s) + \mu r)$, so the value of $\mu r$ limits the effect of $K_{C_0}^A(s_n)$ on

increasing the value of $J_A(\mu, r, s)$. Moreover, in the case of low protein expression is when the value of $\mu r$ is maximized with respect to the case of high protein expression, as there is less competition for recruiting ribosomes.

At high protein expression, both the promoter and RBS strength determine the TRY value. Figure 3A shows that the high protein expression region ($\mu \in [0, 0.015]$ min$^{-1}$) corresponds to medium to high values of $N_A \omega_A$ (yellow and red dots), as we discussed in the previous paragraph. Nevertheless, in contrast to the low protein expression case, the value of $K_{C_0}^A(s_n)$ does influence the value of the TRY measures (size of the yellow and red dots). In particular, high values of $N_A \omega_A$ and $K_{C_0}^A(s_n)$ (big red dot) maximizes the values of titer and yield. Otherwise, balanced combinations of $N_A \omega_A$ and $K_{C_0}^A(s_n)$ (medium yellow dot or small red dot) maximize the value of productivity rate.

In summary, on the one hand, increasing the promoter strength makes the value of $J_A(\mu, r, s)$ rise proportionally, increasing the protein $A$ expression level (if there are cellular resources that can be recruited by increasing the value of $J_A(\mu, r, s)$). On the other hand, increasing the RBS strength will not always increase $J_A(\mu, r, s)$ since its effect is limited by the value of $\mu r$. Thus, the promoter strength mainly determines the TRY values, and the RBS strength can adjust it with some limitations. Finally, there are combinations of different promoter and RBS strength values that result in the same protein expression value for a given substrate, which what we focus on in the next section.

### 3.3. TRY Relative Variation Indices Are Fundamentally Different at Low and High Cell Burden

Different $N_A \omega_A$ and $K_{C_0}^A(s_n)$ configurations can result in the same value of $J_A(\mu, r, s)$ for a given substrate, and hence the same TRY values. Equation (9) suggests these different configurations could have different sensitivity to variations in $s$ and $\mu r$. To test this, we used the sensitivity indices defined in (22) to analyze how much the TRY space varies as a function of variations in the substrate availability for different configurations in the expression space.

Figure 3B shows that, at low protein expression ($\mu \in [0.015, 0.025]$ $min^{-1}$), the relative variation indices $RV_\chi$ span a wide range of values, from being close to zero (little or no variation in TRY measures due to variations in substrate) to values greater than 1.5 (large variation in TRY measures due to variations in substrate). This result confirms that, at low protein expression, different configurations of $N_A \omega_A$ and $K_{C_0}^A(s_n)$ have different sensitivities in variations of $s$.

However, Figure 3B shows that increasing protein expression reduces the range of possible $RV_\chi$ values. In particular, at high protein expression, $\mu \in [0, 0.005]$ $min^{-1}$, all $N_A \omega_A$ and $K_{C_0}^A(s_n)$ configurations converge to the same $RV_\chi$ values. Specifically, $RV_T$ and $RV_Y$ approach to zero, while $RV_R$ approaches to 0.5. This result shows that, at high protein expression, it is indifferent the $N_a \omega_a$ and $K_{C_0}^A(s_n)$ configuration; all $N_a \omega_a$ and $K_{C_0}^A(s_n)$ configurations tend to the same sensitivity with respect to substrate.

The TRY relative variation indices are fundamentally different at low and high cell burden. This is because in the case of high protein expression, $\mu r$ becomes negligible with respect to $d_m^A / K_{C_0}^A(s)$, so that the $J_A(\mu, r, s)$ equation can be simplified into

$$J_A(s) = 0.62 \frac{l_p^A}{l_e} \frac{\omega_A}{d_m^A} K_{C_0}^A(s). \tag{23}$$

Then, for high protein expression, $J_A(s)$ depends only on the value of $s$, and it is independent of $\mu r$. This explains why at a high level of expression all the combinations of $N_A \omega_A$ and $K_{C_0}^A(s_n)$ tend to the same $RV_\chi$ values; by simplifying Equation (9) into Equation (23), the effect of $N_A \omega_A$ and $K_{C_0}^A(s)$ in the sensitivity of $J_A(s)$ becomes similar.

### 3.4. There Exists a Trade-Off in the Relative Variation Indexes

In the previous sections, we have shown how $N_A \omega_A$ and $K_{C_0}^A(s_n)$ determine trade-offs in the measures of titer, productivity rate and yield; in this section we investigated

whether there is also a trade-off in the relative variation indexes and how $N_A\omega_A$ and $K_{C_0}^A(s_n)$ affect it. Next, we show which conditions are necessary to minimize each of the relative variation indexes.

The titer and yield relative variation indexes are zero when the mass fraction between host cells and protein $A$ is constant with changes in substrate. The titer and yield equations are independent of growth rate; therefore, for their relative variation indexes to be zero, it is sufficient for the cell to invest the same mass fraction as protein $A$ regardless of substrate changes.

The productivity relative variation index is zero when the product of growth rate by the fraction of mass invested as protein $A$ is constant with changes in substrate. The productivity equation depends on the growth rate; then, for its index to be zero, it is not sufficient to achieve a constant mass fraction invested as protein $A$ (as was the case with titer and yield). To achieve a productivity relative variation index of zero, it is necessary that the product of mass fraction and growth rate remains constant.

Therefore, there exists a trade-off between the titer and yield indexes versus the productivity index since they require different conditions to minimize their values that are incompatible with each other. Figure 3B shows that none of the combinations of the colored dots is able to make all three indices zero at the same time (this also true for the gray dots, although it is not shown in the graph).

To take a deeper look into the relative variation index $RV_\mathcal{X}$ and their meaning, it is necessary to analyze the variation of indices for changing levels of substrate. Figure 4 shows that a change in the substrate can affect the TRY values by causing them to go down or up with respect to the TRY nominal value (it may even be the case that the TRY has one section where TRY goes up and another where TRY goes down). For example, Figure 4B shows that in the case of titer it is possible to select a value of $k_u$ (solid blue line) that allows us to increase the titer when the substrate decreases, and we can also select a different value of $k_u$ (dotted blue line) that decreases the titer when the substrate decreases.

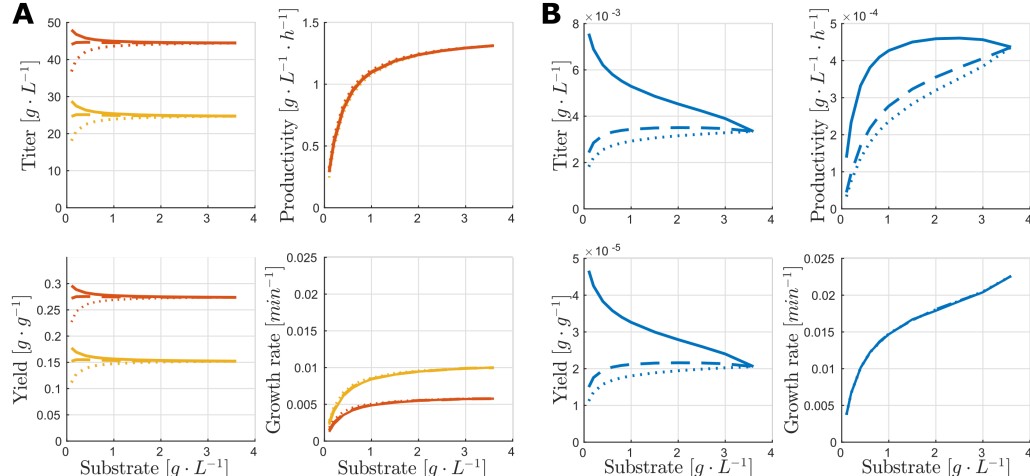

**Figure 4.** TRY at different nominal substrate concentrations for high and low protein expression levels for fedbatch operation. Orange and yellow (high and medium burden) lines correspond to different $N_A\omega_A$ and $K_{C_0}^A(s_n)$ combinations that achieved the same productivity value of 1.31 g/L/h at the nominal substrate 3.6 g/L, while blue (low burden) lines correspond to combinations that achieved $4.3 \times 10^{-4}$ g/L/h. We chose $k_b = 15$ for all the combinations, $k_u = \{6, 20, 135\}$ for orange and yellow, $k_u = \{6, 80, 135\}$ for blue, and we used $N_A\omega_A$ to achieve the same productivity level. Low levels of $k_u$ are drawn with dotted lines, mid levels with dashed lines and high levels with solid lines. Then, for these combinations, we reduced the nominal substrate up to 0.1 g/L to see its effect on titer, productivity rate, yield and growth rate. (**A**) TRY for medium and high burden $N_A\omega_A$ and $K_{C_0}^A(s_n)$ combinations. (**B**) TRY for low burden $N_A\omega_A$ and $K_{C_0}^A(s_n)$ combinations.

From this, we can see that both the absolute variation and the direction of change in production are important. For industrial production purposes it is not the same for production to go down than for production to go up with respect to the nominal value. Then, there may be different combinations of $N_A \omega_A$ and $K_{C_0}^A (s_n)$ that achieve the same rate of change, but they may have different directionality, and then one combination can be more favorable than the other. This highlights the importance of the selection process for the values of $N_A \omega_A$ and $K_{C_0}^A (s_n)$.

## 4. Conclusions

In this work, we demonstrate the need for metabolic burden models as well as their utility. Our results show that, at low expression levels, gene transcription mainly defined TRY, and gene translation had a limited effect; whereas, at high expression levels, TRY depended on the product of both, in agreement with experiments in the literature [5]. Our model can be used to predict cell growth rate and cell mass distribution between enzymes of interest and host enzymes as a function of substrate uptake and the main lab-accessible gene expression-related characteristics: promoter strength, gene copy number and ribosome binding site strength. Multiscale models, like the one presented here, incorporating the dynamics of (i) the cell population in the bioreactor, (ii) the substrate uptake and (iii) the interaction between the cell host and the expression of enzymes of interest, are useful to understand the differential roles of gene transcription and translation in shaping TRY trade-offs for a wide range of expression levels and the sensitivity of the TRY space to variations in substrate availability. Optimal gene expression is central for the development of both bacterial expression systems for heterologous protein production, and microbial cell factories for industrial metabolite production. With our approach it will be easier to fulfill industry-level overproduction demands optimally, as measured by the key performance metrics: titer, productivity rate and yield (TRY).

**Author Contributions:** Conceptualization, F.N.S.-N., Y.B., A.V. and J.P.; methodology, F.N.S.-N. and Y.B.; software, F.N.S.-N.; validation, F.N.S.-N., Y.B. and A.V.; formal analysis, Y.B., A.V. and J.P.; writing—original draft preparation, F.N.S.-N. and J.P.; writing—review and editing, Y.B., A.V. and J.P.; visualization, F.N.S.-N. and Y.B.; supervision, A.V. and J.P.; project administration, A.V. and J.P.; funding acquisition, J.P. All authors have read and agreed to the published version of the manuscript.

**Funding:** This research was partially supported by grants MINECO/AEI, EU DPI2017-82896-C2-1-R 662 and MICINN/AEI, EU PID2020-117271RB-C21. F.N.S.-N. thanks the UPV grant number PAID-01-2017.

**Institutional Review Board Statement:** Not applicable.

**Informed Consent Statement:** Not applicable.

**Data Availability Statement:** The open source implementation of the multiscale model and the scripts we used to generate the figures of this work are available at https://github.com/sb2cl/MDPI2021-MultiscaleModel (accessed on 31 May 2021).

**Acknowledgments:** The authors would like to thank Jose Luis Navarro Herrero and Pablo Carbonell Cortés for their ideas and suggestions during the developing of this work.

**Conflicts of Interest:** The authors declare no conflict of interest. The funders had no role in the design of the study; in the collection, analyses, or interpretation of data; in the writing of the manuscript, or in the decision to publish the results.

## Abbreviations

The following abbreviations are used in this manuscript:

TRY     Titer, Rate, Yield
RBS     Ribosome binding site

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
