# Peer review of "Gene Expression Space Shapes the Bioprocess Trade-Offs among Titer, Yield and Productivity"

_applsci, doi:10.3390/app11135859_

Round 1

Reviewer 1 Report

The authors represent a mathematical model to evaluate metabolic burden throughout fed-batch, batch, and continuous fermentation. Overall, the model is described well. Furthermore, the abstract reasons the need for a metabolic burden model well. The introduction provides sufficient background for the here presented research. However, there are some minor aspects within the text where further detail would aid understanding.

The authors refer in a number of places to the enzyme of interest (e.g. line 6, 7, 74). This raises the question whether the model only applies to enzymes or to proteins in general. This subject should be addressed and clarified.

The main body of work is phrased well and represents the work clearly. The parameters of the model are shown in Table 2 (Appendix) and mentioned in the figure legend Figure 4. A minor suggestion to the main body of work is that a further representation of the parameters chosen within the text would be aiding the understanding of the reader. Therefore, I would suggest showing Table 2 (Appendix) in the Materials and Methods instead.

The authors refer to the enzyme cost (e.g. line 43). Here, it would be good to specify if the cost relates to financial costs or costs for the cell to avoid confusions.

Please clarify the sentence in line 69 to 71. Transcription and translation are frequently looked at in the context of protein expression, however, not modelled. Therefore, this sentenced should be specified.

Furthermore, it would be interesting which E. coli strain has been used for the modelling (line 178), as there are differences in expression abilities and growth rates of different E. coli strains.

The authors should state clearly that the data created is based on a model and not validated with real-time expression data. Furthermore, the authors only show one application of the model for one protein. I strongly recommend to show data for a further protein to validate the model. Additionally the here used protein is called ‘protein A’ which has not been referenced. Please clarify whether ‘protein A’ is an E. coli native protein, a recombinant protein, or a theoretical protein. If it is a native protein or a recombinant protein, it would preferred to reference it.

For further publications, the overall merit could be improved showing a data from a protein expression experiment which would provide evidence for a wide application area. Therefore, I would highly recommend to use biological experiments in the future. Also, as no biological experiments are shown, discussing the validity of the model in relation to experimental data and how the model can be transferred on a biological experiment, would be crucial.

Furthermore, I would like to mention some minor aspects about style and grammar:

  1. No abbreviations in the abstract are preferred
  2. Sentence in line 84 should be deleted as it does not aid readability
  3. Figure 1A: missing spacebar behind the A
  4. Figure 1: In the current presentation, it is slightly misleading which figures are B and C. Some separation would aid understanding
  5. Equation 1 should be separate Equations 1 to 3
  6. Line 113: Style of referencing appears to have changed
  7. Line 213: non-monotonously
  8. Line 233 to line 235: Sentence is not quite clear. Should it be ‘[…] as in much modelling this provides […]’?
  9. Figure 2: Please adapt the figure legend that it states that these data is based on a model to avoid misunderstanding
  10. Figure 3: The legend contains an additional z at the beginning of the legend
  11. Line 251: equation should be spelled with a small e and to maintain the same style the 7 should be in brackets
  12. Line 338, please, when mentioning literature references, also reference which literature is referred too
  13. It would be good to show figure 4 before the conclusion
  14. Reference 10 contains an additional hyperlink
  15. Reference 30 contains an additional hyperlink
  16. Reference 42 – no authors are named, these should be added

Author Response

Thank you very much for your comments. They were very helpful to improve the paper.
Next we address the concerns of each reviewer. We use blue color in the manuscript to show the new changes.

Q1. The authors refer in a number of places to the enzyme of interest (e.g. line 6, 7, 74). This raises the question whether the model only applies to enzymes or to proteins in general. This subject should be addressed and clarified.
A1. Thank you for the comment, indeed the approach works for any protein, enzymes being of interest for metabolite production. We have changes in the appropriate places where now we refer to proteins in general.  

Q2. The main body of work is phrased well and represents the work clearly. The parameters of the model are shown in Table 2 (Appendix) and mentioned in the figure legend Figure 4. A minor suggestion to the main body of work is that a further representation of the parameters chosen within the text would be aiding the understanding of the reader. Therefore, I would suggest showing Table 2 (Appendix) in the Materials and Methods instead.
A2. Thank you for the comment. The corresponding tables were moved to the Methods sections. 

Q3. The authors refer to the enzyme cost (e.g. line 43). Here, it would be good to specify if the cost relates to financial costs or costs for the cell to avoid confusions.
A3. We understand the possible confusion, we have modified in the text, e.g. line 42-44 now reads: "Models that integrate metabolism, biomass composition and the cost for the cell to produce a certain protein or enzyme can potentially yield better predictions than those focused on metabolism in isolation"

Q4. Please clarify the sentence in line 69 to 71. Transcription and translation are frequently looked at in the context of protein expression, however, not modeled. Therefore, this sentenced should be specified.
A4. Thank you for the comment, we have clarified this sentence.

Q5. Furthermore, it would be interesting which E. coli strain has been used for the modelling (line 178), as there are differences in expression abilities and growth rates of different E. coli strains.
A5. We included the appropriate references, and now we mention that the parameters were fitted using experimental data from the literature, for E. coli K-12 strain MG1655 and E. coliB/r strain.

Q6. The authors should state clearly that the data created is based on a model and not validated with real-time expression data. Furthermore, the authors only show one application of the model for one protein. I strongly recommend to show data for a further protein to validate the model. Additionally the here used protein is called ‘protein A’ which has not been referenced. Please clarify whether ‘protein A’ is an E. coli native protein, a recombinant protein, or a theoretical protein. If it is a native protein or a recombinant protein, it would preferred to reference it.
A6. We understand the possible confusion, we stated now at the beginning (lines 115-116): "we consider the expression of a generic theoretical gene A". As stated in the fisrt answer, our approach works for any protein provided one can specify the length pf the protein in aminoacids. In our example, Protein A has 195aa (this is the value of the parameter l_p^A in the new Table 3).

Q7. For further publications, the overall merit could be improved showing a data from a protein expression experiment which would provide evidence for a wide application area. Therefore, I would highly recommend to use biological experiments in the future. Also, as no biological experiments are shown, discussing the validity of the model in relation to experimental data and how the model can be transferred on a biological experiment, would be crucial.
A7. Thank you for the advice. We are working on a experimental publication where we use our model to design and predict the productivity of a given protein, and the show the experiments give us those results. The host model, however was calibrated using experimental data from the literature as explained now in Q5. 

All the minor aspects were addressed in the manuscript, however, we maintained the abbreviation of TRY in the abstract as is a known abbreviation in the area and explicitly writing tite productivity rate and yield many times would render the text more difficult to read.

Reviewer 2 Report

In the paper entitled “Gene expression space shapes the bioprocess trade-offs among titer, yield and productivity”, the authors presented the metabolic burden model and its need for yield prediction. I think this model might be helpful to predict yield in the industrial-scale production of essential metabolites.  However, I recommend this paper for publication after addressing the below-mentioned issues.
I found an unnecessary sentence end of the introduction – It should be removed.
In figure legend three mentioned as “z TRY”, – it should be clarified 
The authors did not follow the journal’s Instructions for Authors to format the reference list; it should be formatted according to its instructions.

Author Response

Thank you for your comments, we addressed them in the manuscript.